# Antibiofilm Activity of *Amomum tsaoko* Essential Oil on *Staphylococcus aureus* and Its Application in Pork Preservation

**DOI:** 10.3390/foods14040662

**Published:** 2025-02-15

**Authors:** Zhifeng Yan, Junrui Guo, Qiming Chen, Sibao Wan, Zhen Qin, Haiyan Gao

**Affiliations:** School of Life Sciences, Shanghai University, Shanghai 200444, China; yzflxt@shu.edu.cn (Z.Y.); guoruirui@shu.edu.cn (J.G.); cqm2016@shu.edu.cn (Q.C.); wansibao@126.com (S.W.)

**Keywords:** biofilms, *Amomum tsaoko* essential oil, antibiofilm activity, pork preservation

## Abstract

*Staphylococcus aureus* (*S. aureus*) can contaminate food by forming biofilms, leading to significant food safety concerns. *Amomum tsaoko* essential oil (AEO) has been shown to be an effective plant-derived antibacterial agent. This study investigated the antibiofilm activity of AEO and evaluated its potential benefit in pork preservation. The results showed that AEO solution (2 mg/mL) can effectively remove the biofilm of *S. aureus* on food contact materials, achieving a removal rate of over 90%. Scanning electron microscopy (SEM) revealed that the *S. aureus* biofilm structure was disrupted after treatment with AEO. Meanwhile, AEO treatment significantly reduced the initial formation of *S. aureus* biofilms and extracellular polymeric substance (EPS) production. In addition, AEO down-regulated the expression of key biofilm-associated genes, including *icaA*, *icaB*, *agrA*, *cidA*, *cidB*, and *cidC*, thereby regulating formation. AEO also exhibited significant antibiofilm activity in pork preservation, effectively controlling key indicators associated with pork spoilage. This study revealed the potential of AEO in food preservation, demonstrating its ability to disrupt *S. aureus* biofilms by inhibiting initial formation, reducing the release of EPS secretion, and regulating the expression of biofilm-associated genes.

## 1. Introduction

*Staphylococcus aureus* (*S. aureus*) is a foodborne pathogen that not only causes food contamination but also poses potential risks to human health. It is frequently found in protein-rich foods like poultry products, dairy products, and pastry products, contributing to food contamination [1,2,3,4]. The bacterium *S. aureus* can survive in food and even adhere to surfaces of food processing equipment, such as stainless steel, glass, and plastic, where it forms biofilms [5]. Biofilms consist of a dense network structure of self-produced autocrine EPSs [6]. Compared to planktonic bacteria, biofilm-forming cells are more resistant to removal. Biofilm has become a problematic matter across various food industries, such as beer production, fish processing, and pork production [7,8]. Consequently, reducing the hazards posed by biofilms has become an important goal for the food industry.

Methods to prevent biofilm contamination include physical sterilization and chemical preservatives [9,10]. Nevertheless, in everyday applications, physical sterilization methods are not appropriate for most types of foods because of the significant equipment requirements. Additionally, the environmental issues and the potential for bacterial resistance linked to the use of chemical preservatives to combat biofilm cannot be overlooked [11]. Therefore, environmentally friendly plant extracts are gaining recognition as promising antibiofilm agents [12,13]. Alongside certain polysaccharides and polyphenols extracted from plants, the antibiofilm activity of plant essential oils (EOs) has been widely studied [14]. EOs are intricate blends of volatile compounds, predominantly obtained through hydrodistillation from aromatic plants [15]. They mainly consist of terpenoids, hydrocarbons and phenols, which frequently demonstrate potent antibacterial and antioxidant activities [16,17]. Previous studies have found that oregano and *Litsea cubeba* can significantly reduce the amount of *S. aureus* [18,19], while cinnamon essential oil can significantly inhibit *Pseudomonas aeruginosa* [20]. EOs have been identified as a potential natural preservative source, and have garnered significant attention from researchers.

*Amomum tsaoko* essential oil (AEO) is a viscous liquid derived from the fruit of *Amomum tsaoko*, which is commonly found in southwest China and is recognized as a popular commercial spice utilized as a food additive [21]. AEO has demonstrated high efficiency against various foodborne pathogens and spoilage microorganisms. AEO exhibits inhibitory effects against both Gram-positive bacteria and Gram-negative bacteria [22]. The antibacterial activity of AEO against bacteria is primarily attributed to its ability to disrupt cell membrane integrity and alter cellular morphology. The leakage of alkaline phosphatase from *Listeria* monocytogenes treated with AEO indicates that AEO can also disrupt the bacterial cell wall structure [23]. Our previous research identified more than 100 components of AEO through GC-MS, such as 1,8-cineole, *α*-pinene, (E)-dec-2-enal, and citral, which exhibited significant inhibitory effects against *S. aureus* [24]. Notably, AEO demonstrates a notable ability to inhibit *S. aureus* at 0.20 mg/mL, proving more effective than other essential oils [25]. While some studies have examined the antibacterial properties of AEO, its potential antibiofilm activity remain unclear, and the application potential of AEO in food preservation still needs to be further investigated.

This study determined the efficacy of AEO in removing *S. aureus* biofilm. Additionally, the mechanism of antibiofilm activity of AEO was explored by analyzing changes in micromorphology, initial adhesion, EPS content, and the expression of genes associated with biofilm formation in *S. aureus*. Finally, AEO was applied to fresh pork to assess its ability to delay spoilage. This study aims to provide valuable knowledge regarding the potential application of AEO and the creation of novel antibiofilm strategies in food preservation.

## 2. Materials and Methods

### 2.1. Chemicals and Bacterial Culture

Anhydrous sodium sulfate, dimethyl sulfoxide, Tween 80, ethanol, acetone, glutaraldehyde, and formaldehyde were purchased from China National Pharmaceutical Group Chemical Reagent Co., Ltd., in Shanghai, China, and were of analytical grade. Crystal violet, 3-(4,5-dimethyl-2-thiazolyl)-2,5-diphenyl-2-H-tetrazolium bromide (MTT), Luria–Bertani (LB), Trypticase Soy Broth (TSB), and Kana were purchased from Beyotime Biotechnology Inc. in Shanghai, China.

The *S. aureus* strain (ATCC 6538) was obtained from the Food Quality and Safety Control Laboratory at Shanghai University (Shanghai, China) and was cultured in LB liquid medium at 37 °C until it reached the logarithmic growth phase.

### 2.2. Essential Oil Extraction

Dried *Amomum tsaoko* samples were purchased from supermarket (ParknShop, Yunnan, China), and stored at 20 °C, shielded from light. The AEO was obtained via the steam distillation method. Specifically, the fruits of *Amomum tsaoko* were ground into a fine powder and filtered through a 40-mesh sieve. Then, 50 g of the *Amomum tsaoko* powder was added to 400 mL of distilled water and subjected to steam distillation for 4 h. The AEO was filtered using a 0.22 µm organic filter and dried using anhydrous sodium sulfate. The AEO was diluted to the appropriate concentration with 0.1% Tween 80 and solvent containing 1% dimethyl sulfoxide. Finally, the extracted AEO was stored in brown glass bottles at 4 °C for future analysis.

### 2.3. The Biofilm-Forming Ability of S. aureus

The ability to form biofilms was assessed using the crystal violet staining method [26]. Briefly, a bacterial suspension (10^8^ CFU/mL, 40 μL) was added to a 96-well plate that contained 160 μL of sterile TSB medium. This was incubated for 6, 12, 24, 36, and 48 h, respectively. After the incubation periods, each well’s biofilm was treated with 1% (*v*/*v*) crystal violet for a duration of 15 min. After washing with sterile water, an ethanol–acetone solution (4:1, *v*/*v*) was used for decolorization, and the optical density at 570 nm was measured using a multi-mode microplate reader (SpectraMax^®^ iD5, Molecular Devices, Sunnyvale, CA, USA).

### 2.4. The Removal Effect of AEO on S. aureus Biofilm

The removal effect of AEO on the biofilm of *S. aureus* was evaluated using the MTT assay. A suspension of *S. aureus* (10^8^ CFU/mL, 50 μL) was combined with 150 µL of sterile TSB into a 96-well plate and cultivated at 37 °C for 24 h. Subsequently, AEO was added at different concentrations. Kana is a commonly used antibiotic, and was used as a positive control. All groups were placed at 37 °C for 10 h. After staining with MTT, the OD_570nm_ value was measured.(1)$$\mathrm{Removal}\;\mathrm{rate}\;\left(\%\right)=[({\mathrm{OD}}_{\mathrm{Control}}\;-\;{\mathrm{OD}}_{\mathrm{Sample}})/{\mathrm{OD}}_{\mathrm{Control}}]\;\times 100$$

### 2.5. Effect of AEO on S. aureus Biofilm on the Surface of Different Food Container Materials

Biofilms were cultured on sterilized glass sheets, plastic sheets (PET, PC, and PVC), and stainless steel with a size of φ1 cm. The biofilms were then treated with AEO at concentrations of 0.5, 1.0, and 2.0 mg/mL AEO for 8 h. Following treatment, the sheets were subjected to ultrasonication (100 W, 15 min) in phosphate-buffer solution (PBS). Then, the samples were incubated at 37 °C for 24 h and the removal effect was assessed by counting the total number of colonies.

### 2.6. SEM Analysis of S. aureus Treated with AEO

Following Section 2.4, biofilms were cultivated on sterile coverslips (2 × 2 cm^2^) and subsequently treated with AEO (2 mg/mL) for 8 h at 37 °C. A control solution containing 0.1% (*v*/*v*) Tween 80 and 1% (*v*/*v*) dimethyl sulfoxide was utilized. The biofilms were rinsed with PBS and then fixed in a 2.5% glutaraldehyde solution at 4 °C for 24 h. After three washes with PBS, the samples underwent dehydration using a gradient of ethanol (50%, 70%, 80%, 90%, and 100%). Subsequently, critical-point drying was performed, and the samples were immediately coated with a thin layer of gold through sputter-coating. The biofilm was visualized under SEM (SU8010, Hitachi Ltd., Tokyo, Japan) at magnifications of 5000× and 40,000×, respectively.

### 2.7. Impact of AEO on the Initial Adhesion of S. aureus in Biofilm Formation

This was carried out according to the method of Cui et al. [27], with modifications. A suspension of *S. aureus* (10^8^ CFU/mL) was inoculated in sterile TSB medium along with AEO (0.125, 0.25 and 0.5 mg/mL) and cultivated at 37 °C until the absorbance at 600 nm (OD_600nm_) reached 0.5. Afterwards, the culture medium was removed by centrifugation and the cells were washed with PBS and then resuspended in PBS. Each group was incubated at 37 °C for 1 h. The samples were fixed using 25% formaldehyde and stained with crystal violet. The absorbance was subsequently measured at 570 nm to evaluate the adhesion, with the relative adhesion rate calculated according to the following formula:(2)$$\mathrm{Relative}\;\mathrm{adhesion}\;\mathrm{rate}\;(\%)={\mathrm{OD}}_{\mathrm{Treatment}}/{\mathrm{OD}}_{\mathrm{Control}}\;\times \;100$$

### 2.8. Effect of AEO on the Release of EPS

In accordance with Section 2.3, sterile coverslips were immersed in TSB medium containing a *S. aureus* suspension and AEO was added at concentrations of 2.7 as mentioned above for processing. Each group was cultured at 37 °C for 2 d, the culture medium was removed, and the wells were washed three times with sterile PBS to discard floating bacteria. Next, PBS was added and a sonicator used to disrupt the biofilm, applying 100 W of ultrasound treatment for 5 min. The resulting bacterial suspension was centrifuged at 5000 rpm for 10 min, and the supernatant filtered through a 0.22 μm membrane to analyze the extracellular matrix content. Extracellular protein levels were measured with the Coomassie Brilliant Blue method. Extracellular polysaccharides were quantified using the phenol-sulfuric acid method [28], and a colorimetric analysis was conducted at 595 nm. The quantity of extracellular DNA (eDNA) was assessed using a bacterial genomic DNA extraction kit (Beyotime, Shanghai, China), with absorbance readings taken at 260 nm [29].

### 2.9. Effects of AEO on the Relative Expression of Biofilm-Formation-Related Genes in S. aureus

RT-PCR was utilized to evaluate the impact of AEO on the relative expression levels of genes involved intracellular adhesin genes (*ica A*, *ica B*), the auxiliary gene regulation gene (*agr A*), and the programmed cell death system operon genes (*cidA*, *cidB* and *cidC*). The sequences of the primers are listed in Table 1. Following treatment of the *S. aureus* biofilm with different concentrations of AEO according to the method described in Section 2.8, total RNA was extracted using a Bacteria Total RNA Isolation Kit (Yeasen Biotechnology (Shanghai) Co., Ltd., Shanghai, China). The template complementary DNA (cDNA) was synthesized via reverse transcription using the ATGScript^®^ RT Mix for qPCR kit (ATG Biotechnology Co., Ltd., Nanjing, China) (Appendix A). RT-PCR was performed by Hieff^®^ qPCR SYBR Green Master Mix (No Rox) (Yeasen Biotechnology (Shanghai) Co., Ltd., Shanghai, China) (Appendix A), with 16S rRNA acting as the internal reference gene. The relative expression levels of the genes were calculated using the comparative threshold cycle (2^−ΔΔCT^) method.

### 2.10. The Effect of AEO on the Removal of S. aureus Biofilm on Fresh Pork

Fresh pork tenderloin samples were obtained from a local fresh food market in Shanghai, China. Prior to cutting the samples into 2 × 2 × 1.5 cm cubes, the intracellular muscle fat and subcutaneous layers were trimmed away. Each group was irradiated under ultraviolet light (254 nm) for 30 min to eliminate natural microbial colonies, and then placed in a bacterial suspension (10^4^ CFU/mL) at 4 °C for 12 h. Subsequently, the samples were washed with PBS and allowed to dry naturally, after which they were maintained at 4 °C for 48 h to facilitate biofilm formation. The experimental groups were subsequently immersed in AEO at concentrations of 2–10 mg/mL for 30 s. The negative control consisted of a solution containing 0.1% (*v*/*v*) Tween 80 and 1% (*v*/*v*) dimethyl sulfoxide, while a Kana solution (0.05 mg/mL) was used as the positive control. The number of colonies was evaluated every day until the fifth day.

### 2.11. Effect of AEO on the Storage Quality of Fresh Pork

Pork is susceptible to contamination by foodborne pathogens, which can lead to spoilage and deterioration. The pork tenderloin samples were sliced into filets measuring 2 × 2 × 1.5 cm and randomly assigned to seven groups (n = 5). The experimental groups were treated with different concentrations of AEO for 30 s. The samples were stored at 4 °C in the refrigerator for 10 d, with a sample taken every 48 h. The samples were blended with distilled water in a 1:10 (*w*/*v*) ratio. The resulting homogenates were then centrifuged at 6000 rpm for 10 min to obtain the supernatant for subsequent experiments. The total viable count (TVC) was determined using the plate counting technique, while the detection of thiobarbituric acid reactive substances (TBARS) was carried out following the spectrophotometric method. The pH value of the samples was determined with a pH meter (Mettler Toledo Inc., Greifensee, Switzerland) and the microdiffusion method was used to determine the total volatile basic nitrogen (TVB-N) [30].

### 2.12. Statistical Analysis

The experimental results were expressed as mean ± standard deviation, with each experiment conducted in triplicate. The collected data were analyzed using one-way analysis of variance (ANOVA) in SPSS version 22.0, where a *p*-value of less than 0.05 was considered a statistically significant difference. Graphs were generated using Origin 2019 b software (Northampton City, MA, USA).

## 3. Results

### 3.1. The Removal Effect of AEO on Biofilms of S. aureus

The findings demonstrate that *S. aureus* biofilms primarily formed within 0–24 h under the experimental culture conditions (Figure 1A). Therefore, the biofilm culture time for *S. aureus* was established at 24 h for subsequent experiments. Changes in the biofilm after treatment with different concentrations of AEO are shown in Figure 1B. As the treatment time and AEO concentration increased, the number of remaining biofilms consistently decreased. In the low concentration groups (below 0.5 mg/mL), the removal rate of biofilm was similar to that observed with an equivalent concentration of Kana solution (*p* > 0.05). When the concentration of AEO was above 1 mg/mL, AEO showed a significantly better performance compared to the Kana solution (*p* < 0.05). Specifically, the removal rates of biofilm treated with 2 mg/mL and 4 mg/mL AEO treatment groups reached 92.27% and 95.94%, respectively, after 10 h. The results indicated that AEO is effective in removing *S. aureus* biofilm.

### 3.2. The Effect of AEO on the Removal of S. aureus Biofilm on Different Food Container Materials

To demonstrate the feasibility of AEO in food applications, we investigated its removal effect on *S. aureus* biofilm from various food container materials (Figure 1C). The colony count in the negative control group remained stable and was consistent with the results of the blank control, suggesting that the negative control had no removal effect on biofilms. In the AEO treatment groups, the efficacy of biofilm removal was found to be directly proportional to the concentration of AEO applied. Notably, AEO at a concentration of 2 mg/mL demonstrated a significantly higher removal effect on different food contact surfaces, achieving a biofilm removal rate as high as 99.99%, while the control group had a higher content of biofilm. These results indicate that AEO exhibits a strong capacity to eliminate *S. aureus* biofilm from the surfaces of various food container materials, suggesting its potential as an effective biofilm remover on surfaces in food applications.

### 3.3. Effect of AEO on the Micromorphology of S. aureus Biofilm

Research suggests that the biofilm in the control group firmly adhered to the surface of the glass slides, forming a dense biofilm structure, with bacterial cells appearing spherical (Figure 2A,B). In contrast, following treatment with AEO, the distance between bacterial cells increased, and the biofilm became dispersed. Additionally, some bacterial cell surfaces exhibited signs of collapse and roughness (Figure 2C,D). These findings indicate that AEO treatment effectively disrupts the micromorphology structure of *S. aureus* biofilm, achieving a removal effect.

### 3.4. Effect of AEO on the Initial Adhesion of S. aureus Biofilm Formation

Bacterial initial adhesion affects the formation of biofilms. The adhesion rate in the control group was relatively high, while the adhesion rate in the AEO treatment group significantly decreased (*p* < 0.05) (Figure 3A). As the concentration of AEO increased, the relative adhesion rate of *S. aureus* biofilm formation exhibited a continuous decrease. After treatment with 0.5 mg/mL AEO, the initial adhesion of *S. aureus* was effectively suppressed, with the adhesion rate reduced to 8.23%. The above study indicates that AEO can reduce the biofilm capacity of *S. aureus* by influencing its initial adhesion.

### 3.5. Effect of AEO on the Release of EPS

The resilience of biofilms to extreme conditions is largely due to the presence of EPSs, which primarily comprise the extracellular protein, extracellular polysaccharide, and eDNA [31]. As shown in Figure 3B–D, a significant decrease in EPSs was observed in a dose-dependent manner following treatment with AEO at concentrations ranging from 0.125 mg/mL to 0.5 mg/mL (*p* < 0.05). After treatment with 0.5 mg/mL AEO, the EPS levels in the *S. aureus* biofilm were significantly reduced compared to the control group (*p* < 0.05). Specifically, the extracellular protein, extracellular polysaccharide, and eDNA content decreased by 82.99%, 72.91%, and 96.62%, respectively. These findings suggest that AEO effectively inhibits the release of EPSs.

### 3.6. Effects of AEO on Relative Expression of Biofilm-Formation-Related Genes of S. aureus

Bacteria that lack polysaccharide intercellular adhesion (PIA) show a marked reduction in their ability to stick to one another at later stages, ultimately preventing biofilm formation [32]. The production of PIA is regulated by the *ica* operon, which includes the *icaA* and *icaB* gene clusters, as well as the *agrA* gene, all of which are associated with the early stages of biofilm formation (Figure 4A). With increasing concentrations of AEO, the expression levels of these genes significantly decreased (*p* < 0.05). The *cidABC* genes primarily regulate programmed bacterial cell death and influence the release of eDNA [33]. After treatment with 0.25 mg/mL AEO, the relative expression levels of *cidA*, *cidB*, and *cidC* decreased by 81%, 55%, and 72%, respectively. Additionally, AEO treatment down-regulated the expression levels of the *atlA*, *sarA*, and *srtA* genes (Appendix A). These results indicated that AEO would reduce the expression of the *icaA*, *icaB, agrA*, *cidA*, *cidB*, and *cidC* genes, thereby affecting biofilm formation (Figure 4B).

### 3.7. Effect of AEO on the Removal of S. aureus Biofilm on Fresh Pork

The bacterial count on the surface of the pork in the negative control group increased significantly after 5 d of storage at 4 °C (Figure 5A). In contrast, bacterial growth in the AEO treatment group was suppressed, with the level of inhibition corresponding to the concentration of AEO applied. Notably, the biofilm removal rate after treatment with 4 mg/mL AEO was similar to that of the Kana treatment group (*p* > 0.05). These results indicate that AEO can effectively remove *S. aureus* biofilm from the surface of pork.

### 3.8. Effect of AEO on Storage Quality of Fresh Pork

After 10 d of storage, the fresh pork treated with AEO and Kana still retained a reddish color, while the control group had turned pale (Appendix A). The primary indicators used to assess the freshness of pork products include TVC, TBARS, pH, and TVB-N. The results of the TVC analysis are shown in Figure 5B, and demonstrate that the colony counts in various sample groups steadily increased as the storage duration lengthened. However, pork treated with AEO demonstrated a significant inhibition of TVC growth. Notably, the 6 mg/mL AEO treatment group showed comparable inhibitory effects on colony formation to the Kana treatment group (*p* > 0.05). In summary, the application of AEO can effectively delay the changes in the TVC of pork.

TBARS serves as a standard index for assessing lipid oxidative byproducts, mainly MDA [34], as shown in Figure 5C. An increase in MDA levels was observed across all samples as the storage time extended. The inhibitory effect of the positive control was comparable in the 4–6 mg/mL AEO treatment groups. By 10 d of storage, compared to the control group, the MDA content in the AEO treatment group was significantly reduced (*p* < 0.05). These results indicate that AEO effectively inhibits lipid oxidation in pork during prolonged storage.

The pH levels of pork samples treated with AEO showed a more gradual increase compared to the control group (Figure 5D). During the storage process, the negative control group displayed the most rapid rise in pH values of the pork samples. Initially, the inhibitory effect of AEO increased with higher treatment concentrations, but subsequently plateaued. Notably, AEO treatment exhibited stronger inhibition at concentrations above 6 mg/mL compared to the positive control group.

TVB-N was utilized to assess the relative freshness of the pork throughout the storage period (Figure 5E). As storage time increased, the elevation of TVB-N was notably more obvious in the negative control group, whereas the AEO treatment groups significantly suppressed the elevation in TVB-N levels. Specifically, in the 6 mg/mL, 8 mg/mL, and 10 mg/mL AEO groups, the TVB-N content remained below 15 mg/100 g even after 8 d of storage. The above study demonstrates that AEO significantly inhibits the decay of pork, thereby extending its shelf life to 8 d.

## 4. Discussion

*S. aureus* is a prevalent foodborne pathogen that can cause the development of biofilms on food processing surfaces, posing a considerable risk for cross-contamination in foods [35]. Once a biofilm has formed, its unique spatial structure makes it challenging to remove, exhibiting enhanced environmental adaptability. Additionally, the widespread use of chemical preservatives in the food industry has led to an increasing prevalence of antibiotic-resistant bacteria. Therefore, controlling biofilms has become a focal point in food safety research.

This study demonstrated that AEO possesses significant antibiofilm activity against *S. aureus*, effectively removing biofilms from various food container materials, including glass, stainless steel, and plastic. After treatment with 2 mg/mL AEO, the biofilm removal rate on PVC reached an impressive 99.99%, slightly higher than those of many biologically active compounds isolated from plants [36,37]. The aggregation and stacking of colonies can be used to form dense three-dimensional structures known as biofilms [38]. SEM clearly demonstrated that AEO significantly disrupts the structure of the biofilm. Following AEO treatment, a reduction in the number of bacterial cells was observed, along with a dispersion of the biofilm structure. Additionally, some bacterial cell surfaces collapsed and became rough, likely due to the exudation of cellular content. This aligns with earlier studies suggesting that different antibiofilm agents can dismantle the intricate structure of biofilms and break apart microcolonies [39].

Biofilm formation and initial colonization require adhesion to the surface of the substrate. Following the initial bacterial adhesion, bacterial proliferation and EPS production rapidly cause the formation of biofilms [40]. This study explored the potential of AEO to decrease the initial adhesion capacity of *S. aureus*, consequently reducing its biofilm-forming ability. During the cell proliferation stage, the production of EPSs contributes to the integrity of the biofilm structure, which primarily consists of extracellular proteins, extracellular polysaccharides, and eDNA [6]. Extracellular protein is the most abundant substance within biofilm EPSs and plays an important role in biofilm formation and stability [41]. Extracellular polysaccharides have several roles, such as facilitating adhesion to surfaces, building and stabilizing biofilm structures, and safeguarding cells against antibacterial treatments [28]. eDNA, which originates from bacterial lysis and bacterial secretion, is essential for initial attachment and biofilm formation [42]. This study demonstrates that AEO can significantly inhibit the secretion of EPS of *S. aureus*, corroborating the results regarding its initial adhesion capacity, thereby suggesting that AEO influences biofilm formation. This is consistent with conclusions drawn by Zhang et al. [43], which demonstrated that 3,3′-diindolylmethane can prevent the production of EPS by *S. aureus*. These results imply that AEO may inhibit the formation of *S. aureus* biofilms by reducing EPS release, thereby compromising the integrity of the biofilm’s three-dimensional structure.

Biofilm formation can be categorized into early, middle, and late stages, each involving the expression and regulation of numerous genes [44]. This study demonstrated that AEO down-regulates the expression of the *icaA*, *icaB*, *agrA*, *cidA*, *cidB*, and *cidC* genes. The early stages of biofilm formation are divided into two phases: the reversible and irreversible adhesion phases. PIA is crucial for promoting reversible adhesion, while quorum sensing (QS) regulates irreversible adhesion [45]. The *icaA/B* genes are responsible for regulating the transfer of N-acetylglucamide (GlcNAc) to the growing PIA chain and for deacetylating PIA to enhance its adhesion properties [46]. Bacteria deficient in PIA exhibit a marked reduction in their ability to adhere to one another, thereby failing to form biofilms. The QS system is an important signaling pathway for biofilm formation, effectively inhibiting biofilm generation by inhibiting the production of signal molecules and interfering with their transmission [47]. AEO influences biofilm formation by regulating the transcription level of genes within the Agr system. The lipopeptide fengycin has been shown to inhibit *S. aureus* by interfering with the Agr system, which aligns with the results of this study [48]. The *cidA*, *cidB*, and *cidC* genes exhibit similar functions to perforin, mainly regulating programmed cell death, bacterial lysis, and the release of eDNA [49]. The results showed that AEO down-regulates the expression of the *icaA* and *icaB* genes, with the proteins encoded by these genes inhibiting the synthesis of PIA and subsequently affecting biofilm formation. Furthermore, AEO also influences biofilm formation by downregulating the expression of the *agrA* gene. Overall, AEO inhibits the expression of the *icaA*, *icaB*, *cidA*, *cidB*, and *cidC* genes. This also suggests that AEO reduces biofilm activity by inhibiting the expression of genes related to extracellular polysaccharides and extracellular DNA in EPS.

To assess the effective antibiofilm activity of AEO, its application in pork preservation was further investigated. This study verified that AEO can effectively reduce *S. aureus* biofilm on the surface of pork. After 10 d of storage at 4 °C, bacterial growth in the AEO-treated group was significantly inhibited, with the level of inhibition correlating positively with the concentration of AEO used. The TVC, TBARS, pH, and TVB-N are key indicators used to indicate the freshness of pork. The TBARS level, which indicates the extent of lipid oxidation, rises as unsaturated fatty acids in pork samples decompose and oxidize. Variations in pH can occur due to the enzymatic activity of microorganisms and the autolysis of internal enzymes, which lead to the production of nitrogenous compounds. An increase in TVB-N is linked to the breakdown and metabolism of proteins and other non-protein nitrogen compounds [50]. Therefore, the enhancement of these four indicators is inversely proportional to the freshness of the pork. AEO effectively reduced the growth of TVC during pork storage and slowed down the increases in TBARS, pH, and TVB-N values, extending the shelf life of the pork by up to 8 d. In comparison to the negative control group, AEO demonstrated positive effects on TVC, TBARS, pH, and TVB-N. The essential oil solvent used may not be suitable for applications in food. Future efforts could consider combining techniques such as chitosan or nanoemulsion to enhance food safety while further improving its preservation effectiveness. Subsequently, the sensory evaluation results also showed that consumers still have a high acceptance of pork treated with AEO (Appendix A). These findings suggest that AEO holds promising potential for application in food preservation.

## 5. Conclusions

AEO exhibited significant antibiofilm activity, effectively removing *S. aureus* biofilms attached to food materials. SEM revealed that AEO achieved a better removal effect by disrupting the *S. aureus* biofilm structure. AEO influenced the early colonization of bacteria on the surface of the product and dispersed EPSs, thereby affecting the formation of *S. aureus* biofilms. Furthermore, AEO inhibited the transcription levels of genes associated with EPS synthesis (*icaA*, *icaB*, *cidA*, *cidB*, and *cidC*) as well as the quorum-sensing-related gene (*agrA*). In addition, the application experiments demonstrated that AEO could effectively remove *S. aureus* biofilms on the surface of pork and extend its shelf life. This study enhanced our understanding of AEO antibiofilm activity and offers a new green preservation method for its application in food preservation.

## Figures and Tables

**Figure 1 foods-14-00662-f001:**
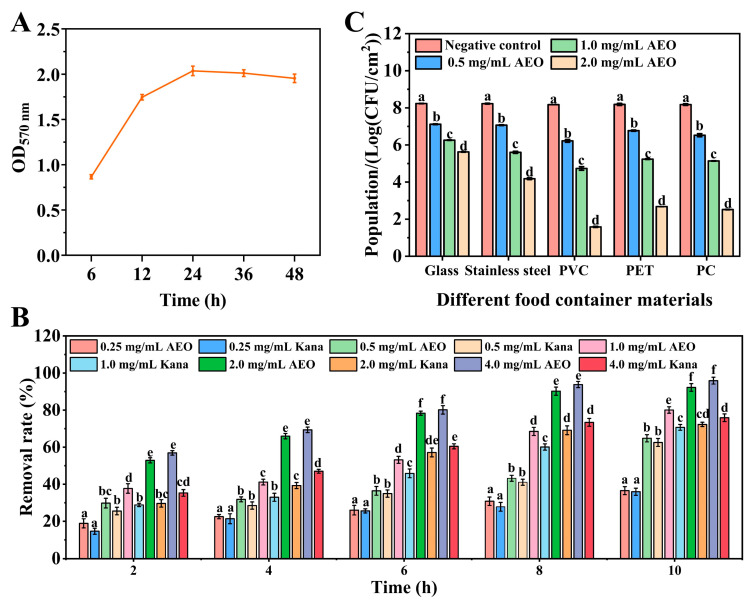
The removal effect of AEO on *S. aureus* biofilm. (**A**) Biofilm formation ability of *S. aureus*. (**B**) The removal effect of AEO on *S. aureus* biofilm on the 96-well plate. Kana solution was used as a positive control. (**C**) Effect of AEO on the biofilm removal for different food materials. A mixture of 0.1% (*v*/*v*) Tween 80 and 1% (*v*/*v*) dimethyl sulfoxide was used as the negative control. The experimental results are presented as the mean ± standard deviation (n = 3). Different letters indicate significant differences between groups (*p* < 0.05).

**Figure 2 foods-14-00662-f002:**
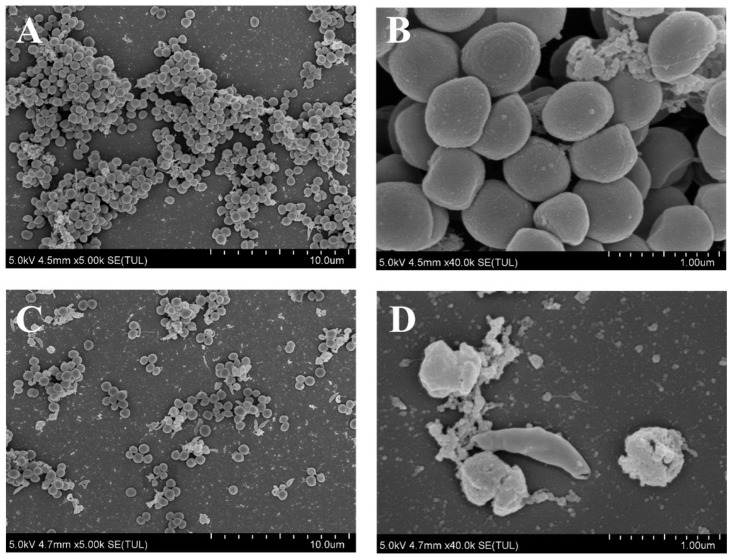
SEM of *S. aureus* biofilm formed on the coverslips. (**A**,**B**) Scanning electron microphotographs of untreated *S. aureus* biofilms. (**C**,**D**) Scanning electron microphotographs of *S. aureus* biofilm treated with AEO (2 mg/mL). Images were taken at 5000× and 40,000× magnification.

**Figure 3 foods-14-00662-f003:**
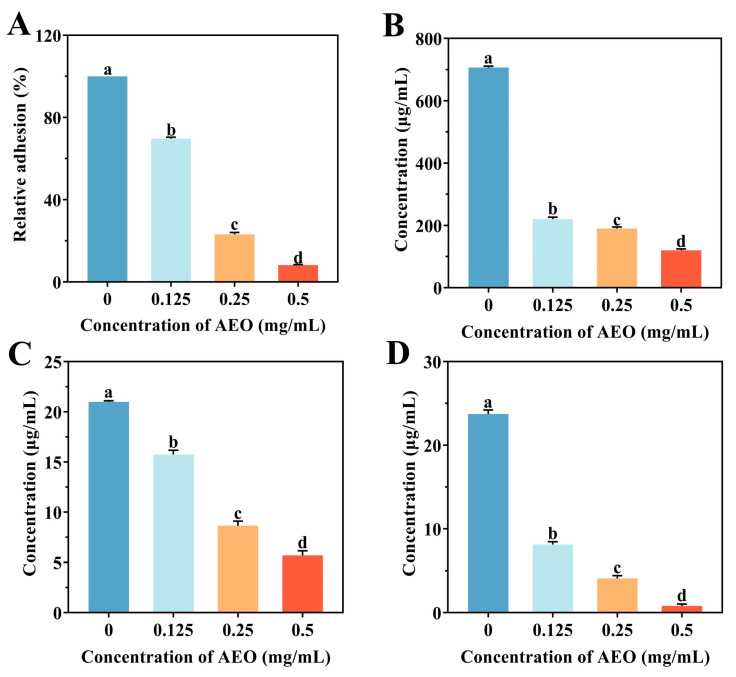
Effect of AEO on biofilm formation. Effect of AEO on initial adhesion (**A**), extracellular protein (**B**), extracellular polysaccharide (**C**), and eDNA (**D**) of *S. aureus* biofilm. A mixture of 0.1% (*v*/*v*) Tween 80 and 1% (*v*/*v*) dimethyl sulfoxide was used as the control (zero point). Different letters indicate significant differences between groups (*p* < 0.05).

**Figure 4 foods-14-00662-f004:**
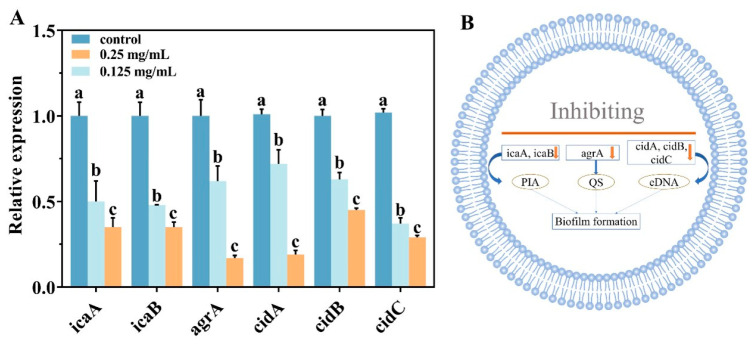
Effect on expression of genes related to *S. aureus* biofilm formation. (**A**) The related gene expression levels as assessed using qRT-PCR. Control: A mixture of 1% (*v*/*v*) dimethyl sulfoxide and 0.1% (*v*/*v*) Tween 80. (**B**) The regulatory process of related genes (orange arrows indicate down-regulation of genes). PIA: Polysaccharide intercellular adhesion; QS: quorum sensing; eDNA: extracellular DNA. Different letters indicate significant differences between groups (*p* < 0.05).

**Figure 5 foods-14-00662-f005:**
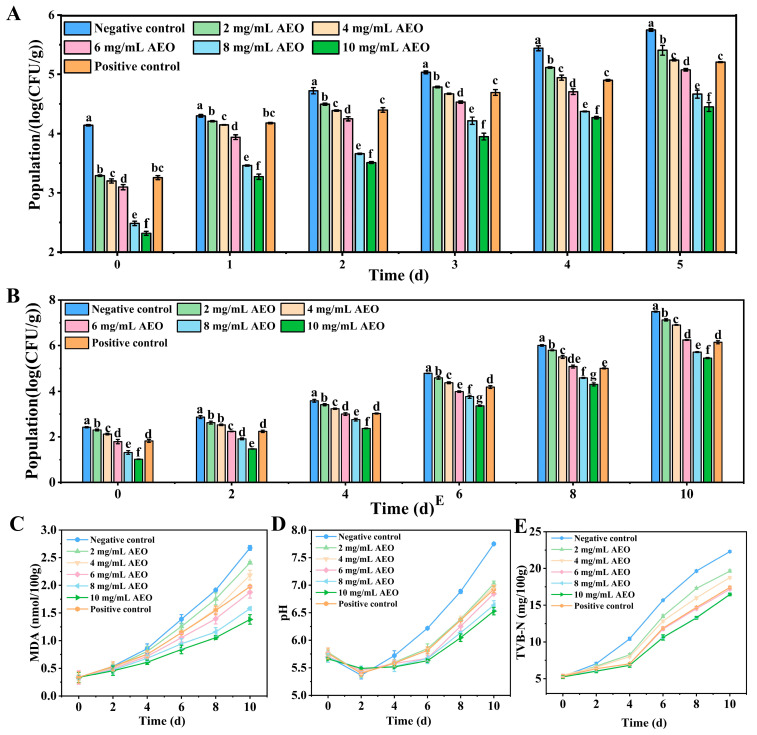
Effect of AEO on fresh pork preservation. (**A**) AEO removal effect on *S. aureus* biofilm on fresh pork surfaces. Effect of AEO on TVC (**B**), TBARS (**C**), pH (**D**), and TVB-N (**E**). All pork samples were stored at 4 °C for 10 d. Negative control: A mixture of 0.1% (*v*/*v*) Tween 80 and 1% (*v*/*v*) dimethyl sulfoxide. Positive control: Kana solution (0.05 mg/mL). Different letters indicate significant differences between groups (*p* < 0.05).

**Table 1 foods-14-00662-t001:** The primers used for RT-qPCR.

Gene	Sequence (5’-3’)
*icaA*	Forward: AGTGCAGTTGTCGATGTTGGCTAC Reverse: ACACATGGCAAGCGGTTCATACTT
*icaB*	Forward: TTTGAAACACATACCCACGATTTGC Reverse: TTGGAGTTCGGAGTGACTGCTT
*agrA*	Forward: CTGATAATCCTTATGAGGTGCTTGA Reverse: AGGTAAGTTCACTGTGACTCGTA
*cidA*	Forward: CTTGGGTAGAAGACGGTGCAAACTT Reverse: AGCGTAATTTCGGAAGCAACATCCA
*cidB*	Forward: AGCCGCAGTAGGTATCGAAGTGT Reverse: CTAGTGCTTTAGCTGTGCCAAATGC
*cidC*	Forward: GGTACAATGGGTTGCGGTCTTCC Reverse: ACCTTTACCACCTGCTGCCTCA
*16S rRNA*	Forward: TGTTCTCAGTTCGGATTGTA Reverse: ATACGGCTACCTTGTTACG

## Data Availability

The original contributions presented in this study are included in the article/Appendix A. Further inquiries can be directed to the corresponding authors.

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
