# Peer review of "Antibiofilm Activity of *Amomum tsaoko* Essential Oil on *Staphylococcus aureus* and Its Application in Pork Preservation"

_foods, 2025, doi:10.3390/foods14040662_

Round 1
Reviewer 1 Report
Comments and Suggestions for Authors
Introduction
The topics must be better linked.
Materials and methods
Storage quality of fresh pork
Color analysis??
Drip loss??
Weight loss??
Results and discussion
Pictures of the stored pork??
Additional Comment: Regarding the revision, the manuscript is interesting. Nevertheless, some points must be revised before any publication.
Indeed, the manuscript presents enough scientific quality and originality to be published.
Nevertheless, as I mentioned in my report, the topics in the introduction should be better linked,
and several quality parameters in the storage of quality fresh pork should be presented to improve the final quality of the manuscript (the storage part was not covered by the authors as it should). All the other points you asked about are mainly covered in the manuscript.
Comments on the Quality of English Language
The English could be improved to more clearly express the research.
Reviewer 2 Report
Comments and Suggestions for Authors
Authors,
However, in this case, I genuinely found the manuscript to be of high quality, with no significant issues in its methodology, conclusions, or presentation. The research addresses a relevant gap in the field, and the authors have provided sufficient data and references to support their findings.
You have articulated your objectives for the Antibiofilm Activity of Amomum tsaoko Essential Oil on Staphylococcus aureus and Its Application in Pork Preservation perfectly. The clarity and relevance of your aims set a strong foundation for the study. However, I would like to suggest the following corrections to enhance your paper:
Abstract: Ok.
Introduction
ok
2. Materials and Methods
Line77 – Please insert the ‘room temperature’. I don’t know what your temperature in your country is.
Line 81, 82, 88, 90, 92, 99 etc. What about the chemicals? Where did you have this? Write like this ‘The reagents used were purchased from Sigma-Aldrich Chemie GmbH, Munich, Germany, and were of analytical grade.’
Results and Discussion
3.1 and 3.2 Incredible results
3.3 Good images
Conclusion: ok
